# $q$-NEURONS: NEURON ACTIVATIONS BASED ON STOCHASTIC JACKSON'S DERIVATIVE OPERATORS

## ABSTRACT

We propose a new generic type of stochastic neurons, called $q$-neurons, that considers activation functions based on Jackson's $q$-derivatives, with stochastic parameters $q$. Our generalization of neural network architectures with $q$-neurons is shown to be both scalable and very easy to implement. We demonstrate experimentally consistently improved performances over state-of-the-art standard activation functions, both on training and testing loss functions.

## 1 INTRODUCTION

The vanilla method to train a *Deep Neural Network* (DNN) is to use the *Stochastic Gradient Descent* (SGD) method (a first-order local optimization technique). The gradient of the DNN loss function, represented as a directed computational graph, is calculated using the efficient backpropagation algorithm relying on the chain rule of derivatives (a particular case of automatic differentiation).

The ordinary derivative calculus can be encompassed into a more general *q-calculus* Jackson (1909); Kac & Cheung (2001) by defining the Jackson's *q-derivative* (and gradient) as follows:

$$D_q f(x) := \frac{f(x) - f(qx)}{(1 - q)x}, \quad q \neq 1, x \neq 0. \tag{1}$$

The $q$-calculus generalizes the ordinary Leibniz gradient (obtained as a limit case when $q \to 1$ or when $x \to 0$) but does *not* enjoy a generic chain rule property. It can further be extended to the $(p, q)$-*derivative* Sadjang (2013); Khan et al. (2018) defined as follows:

$$D_{p,q} f(x) := \frac{f(px) - f(qx)}{(p - q)x}, \quad p \neq q, x \neq 0. \tag{2}$$

which encompasses the $q$-gradient as $D_{1,q}f(x) = D_{q,1}f(x) = D_q f(x)$.

The two main advantages of $q$-calculus are

1. To bypass the calculations of limits, and
2. To consider $q$ as a stochastic parameter.

We refer to the textbook Kac & Cheung (2001) for an in-depth explanation of $q$-calculus. Appendix A recalls the basic rules and properties of the generic $(p, q)$-calculus Khan et al. (2018) that further generalizes the $q$-calculus.

To the best of our knowledge, the *q-derivative operators* have seldom been considered in the machine learning community Xu & Nielsen (2018). We refer to Gouvêa et al. (2016) for some encouraging preliminary experimental optimization results on global optimization tasks.

In this paper, we introduce a meta-family of neuron activation functions based on standard activation functions (e.g., sigmoid, softplus, ReLU, ELU). We refer to them as *q-activations*. The $q$-activation is a *stochastic activation function* built on top of any given activation function $f$. $q$-Activation is very easy to implement based on state-of-the-art Auto-Differentiation (AD) frameworks while consistently producing better performance. Based on our experiments, one should almost always use $q$-activation instead of its deterministic counterpart. In the remainder, we define *q-neurons* as stochastic neurons equipped with $q$-activations.

Our main contributions are summarized as follows:

- The generic $q$-activation and an analysis of its basic properties.
- An empirical study that demonstrates that the $q$-activation can reduce both training and testing errors.
- A novel connection and sound application of (stochastic) $q$-calculus in machine learning.

## 2   NEURONS WITH $q$-ACTIVATION FUNCTIONS

Given *any* activation function $f : \mathbb{R} \to \mathbb{R}$, we construct its corresponding "quantum" version, also called *q-activation function*, as

$$g_q(x) := \frac{f(x) - f(qx)}{1 - q} = (D_q f(x))(x),  \tag{3}$$

where $q$ is a real-valued random variable. To see the relationship between $g_q(x)$ and $f(x)$, let us observe that we have the following asymptotic properties:

**Proposition 1.** *Assume $f(x)$ is smooth and the expectation of $q$ is $E(q) = 1$. Then $\forall x$, we have*

$$\lim_{\mathrm{Var}(q) \to 0} g_q(x) = f'(x)x.$$

$$\lim_{\mathrm{Var}(q) \to 0} g_q'(x) = f'(x) + f''(x)x,$$

*where $E(\cdot)$ denotes the expectation, and $\mathrm{Var}(\cdot)$ denotes the variance.*

*Proof.*

$$\lim_{\mathrm{Var}(q) \to 0} g_q(x) = \lim_{\mathrm{Var}(q) \to 0} \frac{f(x) - f(qx)}{x - qx} x = \lim_{qx \to x} \frac{f(x) - f(qx)}{x - qx} x = f'(x)x.$$

$$\lim_{\mathrm{Var}(q) \to 0} g_q'(x) = \lim_{\mathrm{Var}(q) \to 0} \frac{f'(x) - qf'(qx)}{1 - q} = \lim_{\mathrm{Var}(q) \to 0} \frac{f'(x) - qf'(x) + qf'(x) - qf'(qx)}{1 - q}$$

$$= f'(x) + \lim_{\mathrm{Var}(q) \to 0} \frac{qf'(x) - qf'(qx)}{1 - q} = f'(x) + \lim_{q \to 1} \frac{f'(x) - f'(qx)}{x - qx} qx = f'(x) + f''(x)x.$$

$\square$

Notice that as $\mathrm{Var}(q) \to 0$, the limit of $g_q(x)$ is not $f(x)$ but $f'(x)x$. Thus informally speaking, the gradient of $g_q(x)$ carries *second-order information* of $f(x)$. We further have the following property:

**Proposition 2.** *We have:*

$$D_p(g_q(x)) = \frac{1}{1 - p} D_q f(x) - \frac{p}{1 - p} D_{p,pq} f(x).  \tag{4}$$

*Proof.*

$$D_p(g_q(x)) = \frac{g_q(x) - g_q(px)}{(1 - p)x} = \frac{(D_q f(x)) x - (D_q f(px)) px}{(1 - p)x}$$

$$= \frac{1}{1 - p} D_q f(x) - \frac{p}{1 - p} D_q f(px).  \tag{5}$$

Since $D_q f(px) = \frac{f(px) - f(pqx)}{px - pqx} = \frac{f(px) - f(pqx)}{(p - pq)x} = D_{p,pq} f(x)$, eq. (4) is straightforward. $\square$

By proposition 2, the $p$-derivative of the $q$-activation $g_q(x)$ agrees with the original activation function $f$.

See table 1 for a list of activation functions with their corresponding functions $f'(x)x$, where $\mathrm{sigm}(x) = 1/(1 + \exp(-x))$ is the *sigmoid function*, $\mathrm{softplus}(x) = \log(1 + \exp(x))$ is the *softplus function*,

$$\mathrm{relu}(x) = \begin{cases} x & \text{if } x \geq 0 \\ 0 & \text{otherwise} \end{cases}$$

Table 1: Common activation functions $f(x)$ with their corresponding limit cases $\lim_{\mathrm{Var}(q)\to 0} g_q(x) = f'(x)x$.

| $f(x)$ | $\mathrm{sigm}(x)$ | $\tanh(x)$ | $\mathrm{relu}(x)$ | $\mathrm{softplus}(x)$ | $\mathrm{elu}(x)$ | |
|---|---|---|---|---|---|---|
| $f'(x)x$ | $\mathrm{sigm}(x)(1-\mathrm{sigm}(x))x$ | $\mathrm{sech}^2(x)x$ | $\mathrm{relu}(x)$ | $\mathrm{sigm}(x)x$ | $x$        $x \geq 0$ | $\alpha \exp(x)x$   $x < 0$ |

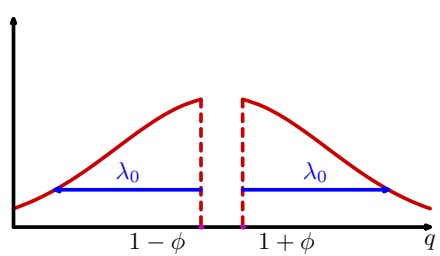 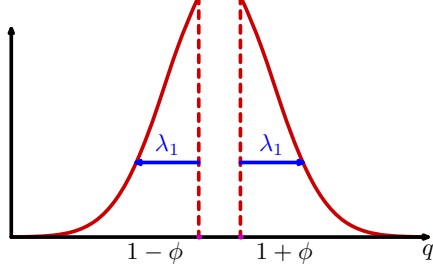

Figure 1: The probability density function of stochastic variable $q$ used when calculating $q$-derivatives.

is the *Rectified Linear Unit* (ReLU) Maas et al. (2013), and

$$\mathrm{elu}(x) = \begin{cases} x & \text{if } x \geq 0 \\ \alpha(\exp(x) - 1) & \text{otherwise} \end{cases}$$

denotes the *Exponential Linear Unit* (ELU) Clevert et al. (2016).

A common choice for the random variable $q$ that is used in our experiments is

$$q = 1 + (2[\epsilon \geq 0] - 1)(\lambda|\epsilon| + \phi), \tag{6}$$

where $\epsilon \sim N(0, 1)$ follows the standard Gaussian distribution, $[\cdot]$ denotes the Iverson bracket (meaning 1 if the proposition is satisfied, and 0 otherwise), $\lambda > 0$ is a scale parameter of $q$, and $\phi = 10^{-3}$ is the smallest absolute value of $q$ so as to avoid division by zero. See fig. 1 for the density function plots of $q$ defined on $(-\infty, -\phi] \cup [\phi, \infty]$

To implement $q$-neurons, one only need to tune the hyper-parameter $\lambda$. It can either be fixed to a small value, e.g. 0.02 or 0.05 during learning, or be annealed from an initial value $\lambda_0$. Such an annealing scheme can be set to

$$\lambda = \frac{\lambda_0}{1 + \gamma(T - 1)}, \tag{7}$$

where $T = 1, 2, \cdots$ is the index of the current epoch, and $\gamma$ is a decaying rate parameter. This parameter $\gamma$ can be empirically fixed based on the total number of epochs: For example, in our experiments, we train 100 epochs and apply $\gamma = 0.5$, so that in the final epochs $\lambda$ is a small value (around $0.02\lambda_0$). We will investigate both of those two cases in our experiments.

Let us stress out that deep learning architectures based on stochastic $q$-neurons are scalable and easy to implement. There is *no* additional free parameter imposed. The computational overhead of $g_q(x)$ as compared to $f(x)$ involves sampling one Gaussian random variable, and then calling $f(x)$ two times and computing $g_q(x)$ according to eq. (3). In our `Python` implementation, the core implementation of $q$-neuron is only in three lines of codes (see A.3).

Alternative approaches to inject stochasticity into neural network training include dropout Srivastava et al. (2014), gradient noise Neelakantan et al. (2016), etc. Both $q$-neuron and dropout modify the forward pass of the neural network. In the experimental section, we will investigate the effect of $q$-neurons with or without dropout. $q$-neuron contracts from the broad array of heristic-based DNN ingredients in that it is based on $q$-calculus, which combines stochasticity and some second-order information in an easy-to-compute way.

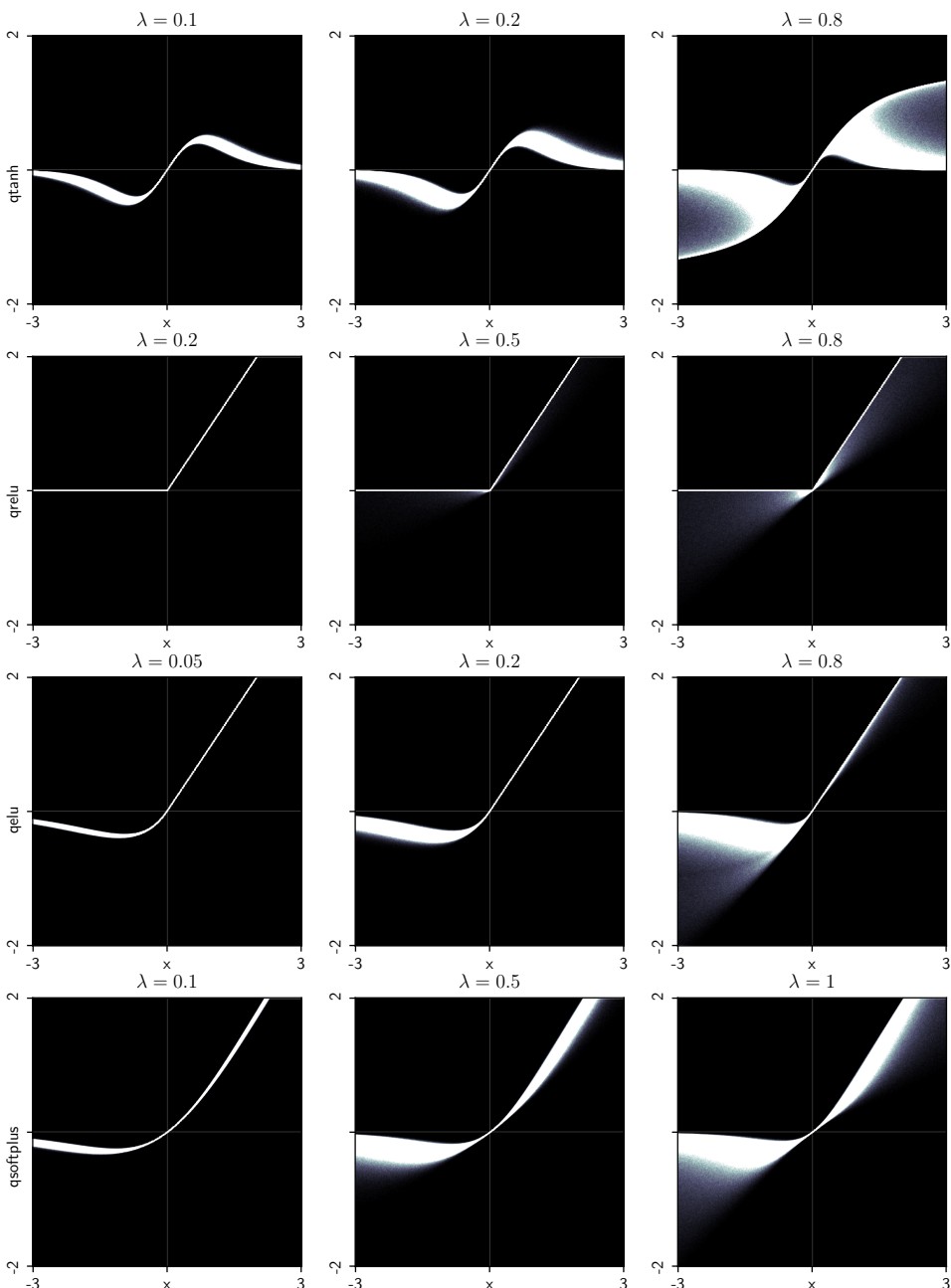

Figure 2: The density function of $q$-neurons with $q$ sampled according to eq. (6) for different values of $\lambda$. The activation is roughly a deterministic function $f'(x)x$ for small $\lambda$ as shown in table 1. The activation is random for large $\lambda$. Darker color indicates higher probability density.

## 3 EXPERIMENTS

We carried experiments on classifying MNIST digits[1] and CIFAR10 images[2] using Convolutional Neural Networks (CNNs) and Multi-Layer Perceptrons (MLPs). Our purpose is not to beat state-of-the-art records but to investigate the effect of applying $q$-neuron and its hyper-parameter sensitivity.

---

[1] http://yann.lecun.com/exdb/mnist/
[2] https://www.cs.toronto.edu/~kriz/cifar.html

The MNIST-CNN architecture is given as follows: 2D convolution with $3 \times 3$ kernel and 32 features; ($q$-)activation; batch normalization; 2D convolution with $3 \times 3$ kernel and 32 features; ($q$-)activation; $2 \times 2$ max-pooling; batch normalization; 2D convolution with $3 \times 3$ kernel and 64 features; ($q$-)activation; batch normalization; 2D convolution with $3 \times 3$ kernel and 64 features; ($q$-)activation; $2 \times 2$ max-pooling; flatten into 1D vector; batch normalization; dense layer of output size 512; ($q$-)activation; batch normalization; (optional) dropout layer with drop probability 0.2; dense layer of output size 10; soft-max activation.

The MNIST-MLP architecture is: dense layer of output size 256; ($q$-) activation; batch normalization; (optional) dropout layer with drop probability 0.2; dense layer of output size 256; ($q$-) activation; batch normalization; (optional) dropout layer with drop probability 0.2; dense layer of output size 10; soft-max activation.

The CIFAR-CNN architecture is: 2D convolution with $3 \times 3$ kernel and 32 features; ($q$-)activation; 2D convolution with $3 \times 3$ kernel and 32 features; ($q$-)activation; $2 \times 2$ max-pooling; (optional) dropout layer with drop probability 0.2; 2D convolution with $3 \times 3$ kernel and 64 features; ($q$-)activation; 2D convolution with $3 \times 3$ kernel and 64 features; ($q$-)activation; $2 \times 2$ max-pooling; (optional) dropout layer with drop probability 0.2; flattern into 1D vector; dense layer of output size 512; ($q$-) activation; (optional) dropout layer with drop probability 0.1; dense layer of output size 10; soft-max activation.

We use the cross-entropy as the loss function. The model is trained for 100 epochs based on a stochastic gradient descent optimizer with a mini-batch size of 64 (MNIST) or 32 (CIFAR) and a learning rate of 0.05 (MNIST) or 0.01 (CIFAR) without momentum. The learning rate is multiplied by $(1 - 10^{-6})$ after each mini-batch update. We compare $\mathrm{tanh}, \mathrm{relu}, \mathrm{elu}, \mathrm{softplus}$ activations with their $q$-counterparts. We either fix $\lambda_0 = 0.02$ or $0.1$, or anneal from $\lambda_0 \in \{1, 5, 9\}$ with $\gamma = 0.5$. The learning curves are shown in figs. 3 to 5, where the training curves show the sample-average cross-entropy values evaluated on the training set after each epoch, and the testing curves are classification accuracy. In all figures, each training or testing curve is an average over 10 independent runs.

For $q$-activation, $c$ means the $\lambda$ parameter is fixed; $a$ means the $\lambda$ is annealed based on eq. (7). For example, "$c0.02$" means $\lambda = 0.02$ throughout the training process, while "$a1$" means that $\lambda$ is annealed from $\lambda_0 = 1$.

We see that in almost all cases, $q$-activation can consistently improve learning, in the sense that both training and testing errors are reduced. This implies that $q$-neurons can get to a better local optimum as compared to the corresponding deterministic neurons. The exception worth noting is $q$-relu, which cannot improve over relu activation. This is because $g_q(x)$ is very similar to the original $f(x)$ for (piece-wisely) linear functions. By proposition 1, $f''(x) = 0$ implies that the gradient of $g_q(x)$ and $f(x)$ are similar for small $\mathrm{Var}(q)$. One is advised to use $q$-neurons only with curved activation functions such as elu, $\mathrm{tanh}$, etc.

We also observe that the benefits of $q$-neurons are not sensitive to hyper-parameter selection. In almost all cases, $q$-neuron with $\lambda$ simply fixed to 0.02/0.1 can bring better generalization performance, while an annealing scheme can further improve the score. Setting $\lambda$ too large may lead to under-fit. One can benefit from $q$-neurons either with or without dropout.

On the MNIST dataset, the best performance with error rate 0.35% (99.65% accuracy) is achieved by the CNN architecture with $q$-elu and $q$-tanh. On the CIFAR10 dataset, the best performance of the CNN with accuracy 82.9% is achieved by $q$-elu.

## 4 CONCLUSION

We proposed the stochastic $q$-neurons based on converting activation functions into corresponding stochastic $q$-activation functions using Jackson's $q$-calculus. We found experimentally that $q$-neurons can consistently (although slightly) improve the generalization performance, and can goes deeper on the error surface.

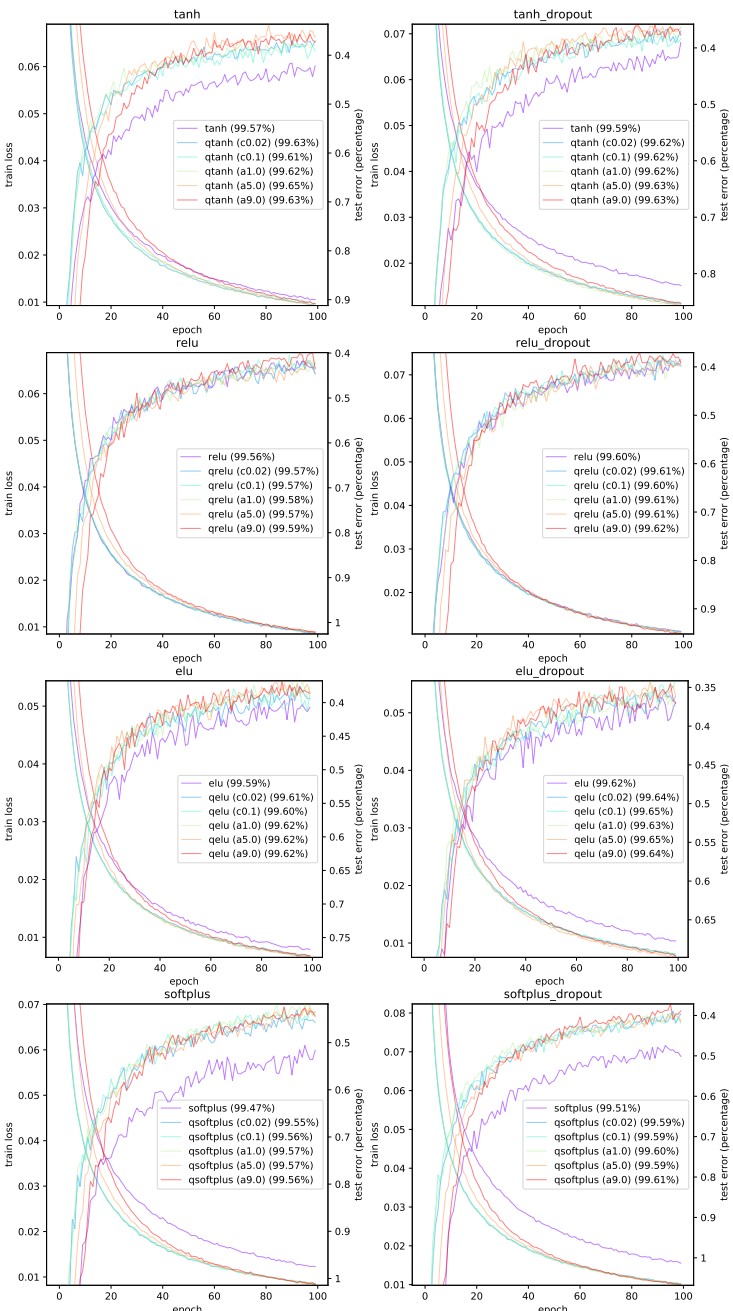

Figure 3: Training loss (descending curves) and testing accuracy (ascending curves) of a CNN on the MNIST dataset, using different activation functions (from top to bottom), with (left) or without (right) dropout.

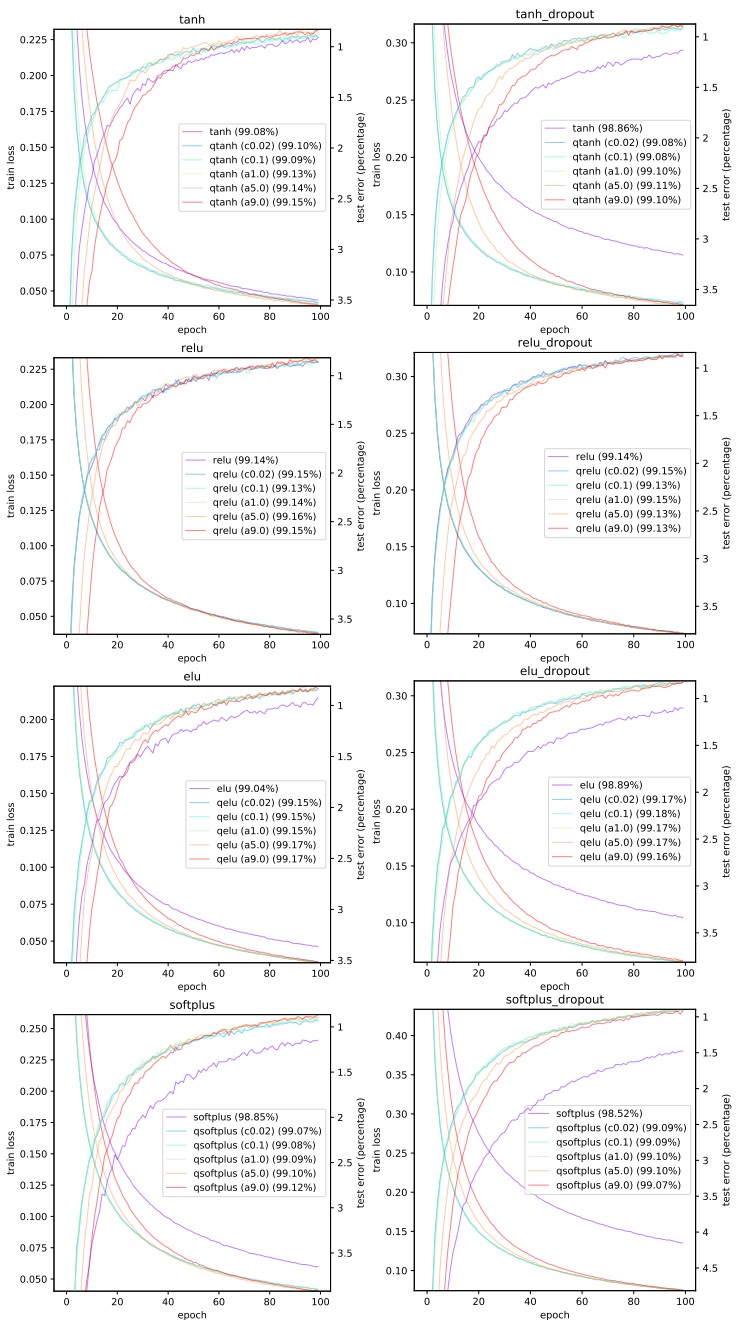

Figure 4: Training loss (descending curves) and testing accuracy (ascending curves) of a MLP on MNIST.

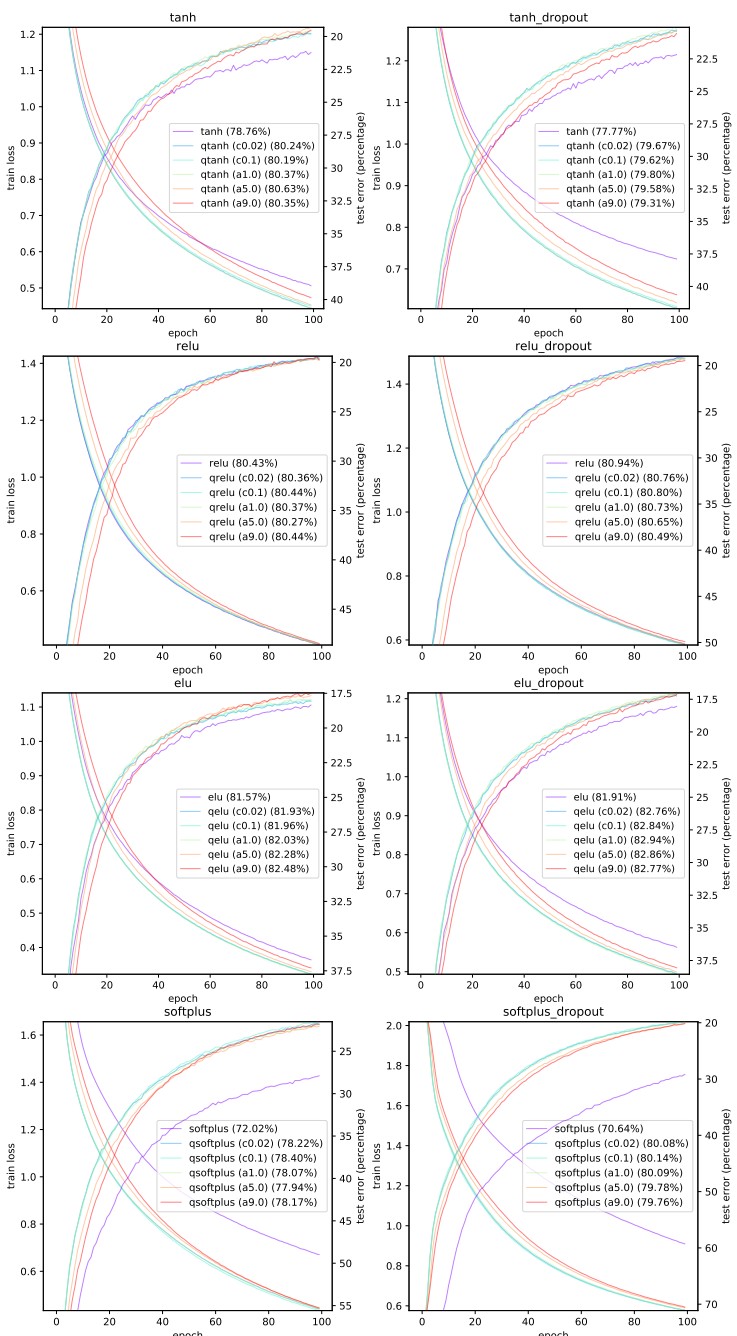

Figure 5: Training loss (descending curves) and testing accuracy (ascending curves) of a CNN on CIFAR10.

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

## A  BRIEF OVERVIEW OF THE $(p, q)$-DIFFERENTIAL CALCULUS

For $p, q \in \mathbb{R}, p \neq q$, define the $(p, q)$-differential:

$$d_{p,q}f(x) := f(qx) - f(px).$$

In particular, $d_{p,q}x = (q - p)x$.

The $(p, q)$-derivative is then obtained as:

$$D_{p,q}f(x) := \frac{d_{p,q}f(x)}{d_{p,q}x} = \frac{f(qx) - f(px)}{(q - p)x}.$$

We have $D_{p,q}f(x) = D_{q,p}f(x)$.

Consider a real-valued scalar function $f(x)$. The differential operator $D$ consists in taking the derivative: $Df(x) = \frac{d}{dx} = f'(x)$.

The $(p, q)$-*differential operator* $D_{p,q}$ for two distinct scalars $p$ and $q$ is defined by taking the following finite difference ratio:

$$D_{p,q}f(x) := \begin{cases} \frac{f(px) - f(qx)}{(p-q)x}, & x \neq 0, \\ f'(0), & x = 0. \end{cases} \tag{8}$$

We have $D_{p,q}f(x) = D_{q,p}f(x)$.

The $(p, q)$-derivative is an extension of Jackson's *q-derivative* Jackson (1909); Kac & Cheung (2001); Ernst (2012); Sadjang (2013) historically introduced in 1909. Notice that this finite difference differential operator that does not require to compute limits (a useful property for derivative-free optimization), and moreover can be applied even to nondifferentiable or discontinuous functions.

An important property of the $(p, q)$-derivative is that it generalizes the ordinary derivative:

**Lemma 3.** *For a twice continuously differentiable function $f$, we have* $\lim_{p \to q} D_{p,q}f(x) = \frac{1}{q}Df(qx) = \frac{1}{q}f'(qx)$ *and* $\lim_{x \to 0} D_{p,q}f(x) = Df(0)$.

*Proof.* Let us write the first-order Taylor expansion of $f$ with exact Lagrange remainder for a twice continuously differentiable function $f$:

$$f(qx) = f(px) + (qx - px)f'(px) + \frac{1}{2}(qx - px)^2 f''(\varepsilon),$$

for $\varepsilon \in (\min\{px, qx\}, \max\{px, qx\})$.

It follows that

$$D_{p,q}f(x) \quad = \quad \frac{f(px) - f(qx)}{x(p-q)} = \frac{f(qx) - f(px)}{x(q-p)}, \tag{9}$$

$$= \quad f'(px) + \frac{1}{2}x(q-p)f''(\varepsilon). \tag{10}$$

Thus, whenever $p = q$ we have $D_{p,q}f(x) = f'(px) = \frac{1}{p}Df(px)$, and whenever $x = 0$, we have $D_{p,q}f(0) = f'(0)$. In particular, when $p = 1$, we have $D_q f(x) = f'(x)$ when $q = 1$ or when $x = 0$. $\qquad\square$

Let us denote $D_q$ the $q$-differential operator $D_q := D_{1,q} = D_{q,1}$.

The following $(p, q)$-Leibniz rules hold:

Since $B_F(qx : px) := f(qx) - f(px) - (qx - px)f'(px) = \frac{1}{2}(qx - px)^2 f''(\varepsilon)$, we can further express the $(p, q)$-differential operator using Bregman divergences Banerjee et al. (2005) as follows:

**Corollary 4.** *We have:*

$$D_{p,q}f(x) \quad = \quad \frac{f(px) - f(qx)}{x(p-q)} = f'(px) + \frac{B_F(qx : px)}{x(p-q)},$$

$$= \quad \frac{f(qx) - f(px)}{x(q-p)} = f'(qx) + \frac{B_F(px : qx)}{x(q-p)}.$$

## A.1 LEIBNIZ $(p, q)$-RULES OF DIFFERENTIATION

- Sum rule (linear operator):

$$D_{p,q}(f(x) + \lambda g(x)) = D_{p,q}f(x) + \lambda D_{p,q}g(x)$$

- Product rule:

$$D_{p,q}(f(x)g(x)) \quad = \quad f(px) + D_{p,q}g(x) + g(qx)D_{p,q}f(x),$$
$$= \quad f(qx) + D_{p,q}g(x) + g(px)D_{p,q}f(x).$$

- Ratio rule:

$$D_{p,q}(f(x)/g(x)) \quad = \quad \frac{g(qx)D_{p,q}f(x) - f(qx)D_{p,q}g(x)}{g(px)g(qx)},$$

$$= \quad \frac{g(px)D_{p,q}f(x) - f(px)D_{p,q}g(x)}{g(px)g(qx)}.$$

General Leibniz rule for $(p, q)$-calculus.

## A.2 THE $(p, q)$-GRADIENT OPERATOR

For a multivariate function $F(x) = F(x_1, \ldots, x_d)$ with $x = (x_1, \ldots, x_d)$, let us define the first-order partial derivative and $i \in [d]$ and $p_i \neq q_i$,

$$D_{p,q,x_i} F(x) \quad := \quad \begin{cases} \frac{F(x_1,\ldots,p_i x_i,\ldots,x_n) - F(x_1,\ldots,q_i x_i,\ldots,x_n)}{(p_i - q_i) x_i} & x_i \neq 0, \\ \frac{\partial F(x)}{\partial x_i} & x_i = 0 \end{cases}$$

$$= \quad \begin{cases} \frac{F(x + (p_i - 1)e_i) - F(x + (q_i - 1)e_i)}{(p_i - q_i) x_i} & x_i \neq 0, \\ \frac{\partial F(x)}{\partial x_i} & x_i = 0 \end{cases},$$

where $e_i$ is a one-hot vector with the $i$-th coordinate at one, and all other coordinates at zero.

The generalization of the $(p, q)$-gradient Soterroni et al. (2013) follows by taking $d$-dimensional vectors for $p = (p_1, \ldots, p_d)$ and $q = (q_1, \ldots, q_d)$:

$$\nabla_{p,q} F(x) := \begin{bmatrix} D_{p_1, q_1, x_1} F(x) \\ \vdots \\ D_{p_d, q_d, x_d} F(x) \end{bmatrix}$$

The $(p, q)$-gradient is a linear operator: $\nabla_{p,q}(aF(x) + bG(x)) = a\nabla_{p,q} F(x) + b\nabla_{p,q} G(x)$ for any constants $a$ and $b$. When $p, q \to 1$, $\nabla_{p,q} \to \nabla$: That is, the $(p, q)$-gradient operator extends the ordinary gradient operator.

## A.3 PSEUDO-CODES

We can easily implement $q$-neurons based on the following reference code, which is based on a given activation function `activate`. Note, q has the same shape as x. One can fix eps$= 10^{-3}$ and only has to tune the hyper-parameter `lambda`.

```
def qactivate( x, lambda, eps ):
    q = random_normal( shape=shape(x) )
    q = ( 2*( q>=0 )-1 ) * ( lambda * abs(q) + eps )
    return ( activate( x * (1+q) ) - activate( x ) ) / q
```

