# OpenReview forum: "q-Neurons: Neuron Activations based on Stochastic Jackson's Derivative Operators"
_ICLR.cc/2019/Conference_

### Official Review · AnonReviewer2 · 2018-11-01
**On the adoption of q-activations**

**Rating:** 5
**Confidence:** 3

**Review:**

The authors describe q-activation functions, stochastic relatives of common activation functions used in neural networks.  It seems like the main argument is to use them because you get a performance improvement with them.

While the experiments appear to show better training at early epochs, none of the models appear to have been trained to convergence.  Additional justifications for why (or when) to use this should be described.

Why does the method outperform particularly when dropout is included?

I also expect the lack of monotonicity in the q-activation functions to lead to the creation of (exponentially) more local minima.  Any comments?

Quality: the experiments need some further work.
Clarity: aside from a few points, the paper is written clearly.
Originality: the work appears original to me
Significance: TBD, but the main argument appears to be that it leads to empirical comparative gains (but on networks not designed to be SOTA).

Small points:
"By prop 2, g_q(x) agrees with with original activation function".  What does "agrees with" mean?
"Fig 2. Darker color --> lighter color?"
"(Conclusion) ... can goes[sic] deeper on the error surface." To me, the experiments only show marginally better performance

---

### Official Review · AnonReviewer1 · 2018-11-01
**My neurons activate unanimously to vote NO for this submission**

**Rating:** 2
**Confidence:** 5

**Review:**


############ Updated Review #################

I have read the author(s)' rebuttal. My decision stays unchanged. In my opinion, this first step is not significant enough, and the presentation is clearly below the acceptance threshold for ICLR. Additionally, the author(s) did not update their submission to reflect the changes. I thereby recommending rejection to this submission.

##########################################

This work proposes to replace the regular deterministic activation functions used in artificial neural nets with stochastic variants. In particular, the author(s) considered the q-derivatives of standard activation functions.

The author(s) claimed that ``By Proposition 2, the p-derivative of the q-activation g_q(x) agrees with the original activation function f.'' I have trouble understanding this. I assume by original activation function the author(s) meant f(x), then how can Eqn (4) agree with f(x)?

At the bottom of pp. 3, the author(s) wrote: ``q-neuron ... combines stochasticity and some second-order information in an easy-to-compute way.'' I definitely can not agree with this point. Basically, q-neuron is the ``derivative'' of the original activation function, so there is no surprise that its derivative links to the second derivative of f(x). I can always use the high order derivative of some function as activation and claim now we are combining even higher order information into the neural network, but does that help? I don't think so.

It really annoys me to see that four out of the eight pages are occupied by gigantic figures, which should be placed in supplementary material in my opinion. A simple table could do the job equally well in the text. We are not interested in nitty-gritty details on how the training evolves. Let alone the datasets tested are all small-scale image classification tasks. At least the author(s) should diversify their test beds (e.g., NLP tasks and ImageNet scale experiments) and model architectures (e.g., RNN, ResNet).

What's also missing from their experiments is a fair comparison with the real counterparts. I do not see comparisons with dropouts, and to more direct activation function randomization schemes (additive noise to regular activation functions).

To summarize, I can not approve this paper as it falls well below the acceptance level of an ICLR. In its current form, it's more like a sketchy note rather than a serious academic paper. I would encourage the authors to significantly enrich the content of this writing before considering resubmitting to another venue.

---

### Official Review · AnonReviewer3 · 2018-11-02
**Interesting theory paper, needs emphasis on usefulness in practice.**

**Rating:** 6
**Confidence:** 3

**Review:**

The authors introduce concept of q-calculus into neural networks along with its advantages. They define a family of stochastic activation functions based on standard functions together with q-calculus.

I have a single question, if the proposed stochastic activation functions can also be achieved through deterministic neurons together with noise schemes like dropout (or any others)? If yes, is it still useful to use q-neurons. sorry, if I missed something obvious.

As mentioned by the authors the experiments only showcase a slight improvement in performance which may not be consistent when tried across larger set of experiments.

---

### Author Response · Authors · 2018-11-26
**Rebuttal**


We would like to thank all reviewers for the helpful comments.
We would like to make the following points to improve the understanding of this work:

-- q-neuron vs deterministic neurons + noise  (reviewer 3)
It is not straightforward to write q-neuron as a deterministic neuron plus noise.
By Figure 2, one can see that the noise has different conditional distributions
wrt the input x and is therefore *not* independent to x.

-- adoption of q-activation (reviewer 2)
Practically, one may use q-neurons for
(1) potentially better performance;
(2) extremely-easy-to-implement noise-injected NN;
Theoretically, q-neuron could be a first step of q-derivative into the deep learning realm.

-- lack of monotonicity leads to more local optima (reviewer 2)
Is it a very interesting direction to explore theoretically and thank you for pointing this out.
We would like to make the following points:
(1) the number of local optima is different from the quality of local optima.
(2) recent developments on deterministic neurons, such as Swish
(Searching for activation functions, Ramachandran et al. 2018)
do not satisfy monotonicity as well.

--proposition 2 (reviewer 1)
Thank you for pointing this out. It should be better phrased as
"The p-derivative of g_q (LHS) agrees with an affine combination of the pq-derivative of f (RHS)."

-- proposition 3 (reviewer 1)
q-derivative is not simply the derivative of the original activation,
It has a profound theoretical background known as q-calculus
(see e.g. Kac and Cheung 2001). We do appreciate your comments
on our empirical evaluation and presentation that we agree to
improve in further versions.

-- dropout (reviewer 1/2/3)
We have already made experimental comparision with/without dropout,
as presented in the left/right of Figure 3/4/5.
Our observation is using q-neuron can bring more performance gains
than dropout.

---

### Meta-Review · Area_Chair1 · 2018-12-15
**Lacking in presentation and in experimental evaluation**

**Confidence:** 4
**Recommendation:** Reject

**Metareview:**

This paper proposes a new type of activations function based on q-calculus. The reviewers found that the papers is significantly lacking in its presentation, in clarity, and in its experimental evaluation. The motivation of the method raises several significant questions to the reviewers, and the proposed method is not sufficiently compared to existing approaches for (noisy) activation functions. After reviews, the authors have failed to present any updates to their paper.